# Knowledge of abortion legality among health facility staff in Ghana

**Grace Sheehy**[1]*, **Chelsea Polis**[1,2], **Easmon Otupiri**[3], **Caroline Moreau**[1]

**1** Johns Hopkins Bloomberg School of Public Health, Baltimore, MD, United States of America, **2** Center for Biomedical Research, Population Council, New York, NY, United States of America, **3** Kwame Nkrumah University of Science and Technology, Kumasi, Ghana

\* sheehyg@ipas.org

**Data Availability Statement:** The dataset and code used for this analysis are publicly available at the following link: https://www.openicpsr.org/openicpsr/project/197801/version/V1/view.

## Abstract

### Background

Abortion has been legal for multiple indications in Ghana since 1985, and efforts have been made to expand the availability of safe abortion care in the years since. However clandestine, and potentially unsafe, abortions remain common, suggesting numerous barriers to access persist; one possible barrier is poor knowledge of the abortion law among those working in health facilities. Our study aimed to identify levels of legal knowledge among health facility staff across Ghana.

### Methods

Data for this paper are drawn from a nationally representative cross-sectional health facility survey conducted in 2018; our analytic sample includes 340 facilities that provide induced abortion and/or postabortion care (PAC). The survey collected data on provision of abortion and PAC, as well as knowledge of abortion legality and recommendations for reducing unsafe abortion. We used descriptive statistics to examine levels of knowledge and recommendations, and logistic regression to assess associations with individual and facility characteristics.

### Findings

Comprehensive knowledge of the legal indications for abortion was low among health facility staff; just 6% identified all legal indications, and the majority (83%) underestimated the number of conditions under which abortion is legal. Knowledge was higher for more restrictive indications, such as a woman's life being at risk, which was identified by 72% of respondents, than more broadly interpretable indications, such as mental health, identified by 29%. Respondents in facilities providing both induced abortion and PAC had better knowledge of several legal indications than those in facilities providing PAC only.

### Conclusions

Health facility staff have significant gaps in their knowledge of abortion legality. Knowledge of the law among this population is highly important for ensuring that abortion care is made

**Funding:** This study was made possible by UK Aid from the UK Government (Component Code: 203177 -101 Purchase Order: 40054781) and a grant from the Dutch Ministry of Foreign Affairs (Activity #4000000282). The findings and conclusions contained within are those of the authors and do not necessarily reflect positions or policies of the donors. The funders had no role in study design, data collection and analysis, decision to publish, or preparation of the manuscript.

**Competing interests:** The authors have declared that no competing interests exist.

available to the fullest extent of the law. Efforts are needed to improve knowledge of the law among providers and facility staff, particularly for indications with broad interpretability.

## Introduction

Ghana has one of the most liberal abortion laws in West Africa. Abortion has been legal for multiple indications since 1985: when a pregnancy poses a threat to life, physical or mental health, and in cases of rape, incest, fetal anomaly, or "defilement of a female idiot" [1,2]. The law further specifies that someone who lacks the capacity to request an abortion can have a surrogate request it for them [3,4]. The Ghanaian government established safe abortion guidelines in 2006 and has made various efforts to expand access to safe abortion care, including expanding the cadre of providers who can perform abortion procedures [3,5,6]. However, clandestine and thus potentially unsafe abortions remain common: in 2017, 71% of abortions in the country were estimated to be illegal (i.e. not performed for current legal indications by registered personnel in an approved facility) [7], and 64% of abortions in the country were estimated to be unsafe [8].

The high incidence of abortion occurring outside the legal framework suggests that those seeking abortion care are not meeting the legal criteria for abortion, and/or that abortion services are not being made available to the fullest extent of the law. This could be due to factors including women being unaware of their legal right to an abortion, providers not being aware of the conditions under which abortion is legal, personal objections to abortion by providers, abortion access being hindered by lack of provider training or supplies, or economic or social barriers, among other reasons. Research among various populations in Ghana has found knowledge of the abortion law is often limited [8,9]. Ensuring abortion care is made available to the fullest extent of the law is a matter of human rights; denying access to legal abortion care constitutes a form of discrimination and a violation of the rights described in the Universal Declaration of Human Rights, including the rights to non-discrimination, life, health, privacy, and bodily autonomy, among others [10,11]. In 2022, the World Health Organization recommended the full decriminalization of abortion due to the adverse impacts of restrictive abortion laws [12].

Abortion laws are often written in unclear or ambiguous terms, with minimal guidance on how they should be interpreted in practice [13]. Specifically, abortion laws often lack information defining the circumstances under which an abortion can be legally obtained; this can have a "chilling effect" on abortion provision, where providers become especially cautious in their interpretation of the law to avoid criminal penalties [13]. Health care providers' knowledge and perceptions of abortion legality can affect their willingness to provide abortion care, thus shaping access considerably. Conversely, legal ambiguity, particularly related to health and social indications for abortion, can also be used to expand access; for instance, in New Zealand, prior to a 2020 amendment, abortion was legal if a pregnancy posed a risk to a woman's mental health and nearly all abortions in the country were granted for mental health indications [14]. In the absence of clear legal guidance, national guidelines, standards and protocols are often written to address this ambiguity and provide practical guidance for health facilities and providers [15]; however, widespread dissemination and uptake of these guidelines is imperative to ensure their utility.

Research from countries including Ethiopia, Ghana, Jamaica, Nepal, and Zimbabwe has demonstrated that knowledge of the national abortion law varies considerably among medical

providers [16–19]. In Zimbabwe, a nationally representative survey of health facilities offering post-abortion care found just 25% of providers knew all four indications under which abortion was legal, while a small number believed abortion was legal for indications not reflected in the law [18]. In Jamaica, 63% of obstetricians and 43% of general practitioners knew the law, and many providers had inaccurate interpretations of the specificities of the law, with many interpreting it more liberally than written [17]. In Nepal, only one-third of providers surveyed in five districts across the country knew all three legal indications for abortion [19]. In a review of Brazilian studies, researchers found that while providers often had adequate knowledge of when abortion was permitted by law, they lacked information on the necessary legal procedures for providing an abortion when a woman's life is at risk, or in the case of fetal anomalies [20]. Less research has explored knowledge of abortion legality among non-medical health facility staff, and most has been concentrated in high-resource settings [21]. However, studies have found that knowledge and perceptions about abortion by health facility management can significantly shape abortion provision at the facility-level [22,23].

A small number of studies have explored knowledge of abortion legality among providers in Ghana. In their survey of providers at Komfo Anokye Teaching Hospital, Morhee et al. (2006) found that 54% were aware that abortion was legal to save a woman's life or preserve her health. Importantly, 76% of respondents stated that the law was not clear enough. Qualitative research has also documented limited knowledge of the abortion law among midwives in rural Ghana [24], and how uncertainty about the legal status of abortion among providers in Accra has led to reluctance to provide abortion care [25]. Martin and colleagues (2011) describe the ambiguity in Ghana's abortion law and how this ambiguity functions as both a deterrent and a facilitator of abortion access–the Ghanaian law states that abortion is unlawful and a punishable offense, and then lists various circumstances under which abortion is legally permitted. This conflicting information creates a sense among some providers that abortion is illegal, breeding anxiety about the possibility of being prosecuted for providing abortion care and contributing to anti-abortion stigma, while offering others the sense that they have more latitude in providing care [26]. In 2006, the Ghana Health Service released guidance for addressing ambiguity in the abortion law via the *Comprehensive Abortion Care Services Standards and Protocols*, recently updated in 2021 [27–29]. The Standards and Protocols provide guiding principles on abortion care service implementation, defining mental health per the WHO definition as a "state of emotional, psychological and social wellbeing and not merely the absence of disease", specifying that continuing a pregnancy "may put a client's mental health at risk" and that no psychiatric assessment is required to meet this indication for a legal abortion [28]. However, wider dissemination of these guidelines is needed to clarify legal ambiguities among providers and health facility staff [25].

To the best of our knowledge, no nationally representative studies have explored knowledge of abortion legality among providers in Ghana. Further, minimal research is available on knowledge of abortion legality among health facility staff more broadly (e.g., administrators). To address this gap, we aimed to:

1. Explore levels of knowledge of abortion legality among staff in non-NGO health facilities offering post-abortion care (PAC) and/or induced abortion across Ghana;

2. Identify health facility staff recommendations for reducing unsafe abortion;

3. Identify respondent- and facility-level factors associated with knowledge of abortion legality and with recommendations for reducing unsafe abortion.

## Methods

### Data source

Data for this analysis come from a nationally representative health facility survey conducted from June through September 2018. The Central Health Information Management System of the Ghana Health Service was used to obtain a list of health facilities that reported data through the District Health Information Management system in 2017. All facility types that could, in theory, provide abortion and/or post-abortion care (e.g., excluding highly specialized facilities such as eye clinics, nutrition centers, and district health directorates), except for community-based health planning and services (CHPS) facilities and non-governmental organizations (NGOs), were retained. The sample frame included 2,758 facilities; complete details on sampling are provided elsewhere [7]. All teaching and regional hospitals were sampled while for lower-level facilities, a two-stage stratified cluster sampling design by facility level was used to generate a nationally representative sample. A total of 539 facilities completed the survey, an 88.7% response rate [7]. Our analytic sample included the 340 facilities who reported providing postabortion care, induced abortion care, or both, since only respondents from these facilities were administered the complete survey, including questions on the conditions under which abortion is legal.

The health facility survey aimed to collect data on annual post-abortion care and induced abortion caseloads [7]. The survey contained five modules, which covered respondent information, facility background and capacity, comprehensive abortion care for legal abortion services, post-abortion care for complications, and general information. The general information module included questions on knowledge of abortion legality in Ghana, perceptions surrounding unintended pregnancy, and recommendations for reducing unsafe abortion.

Interviews were conducted with a senior member of the health facility staff who was knowledgeable about the facility's provision of post-abortion and/or induced abortion care, including obstetrician-gynecologists, general practitioners, physician's assistants, nurses, midwives, and administrators. The questionnaire was administered face-to-face by one of 17 trained interviewers, who were university students and degree-holders with several years of experience conducting reproductive health research. This study received ethical approval from the Johns Hopkins Bloomberg School of Public Health Institutional Review Board (IRB), the Guttmacher Institute's IRB, and the School of Medical Sciences/Komfo Anokye Teaching Hospital Committee on Human Research, Publications and Ethics. The Strengthening the Reporting of Observational Studies in Epidemiology (STROBE) Statement was used to guide reporting on this study (S1 File).

### Measures

**Dependent variables.**   Our primary outcome was knowledge of the legal conditions for abortion in Ghana, assessed via the question "Under what conditions in Ghana is abortion legal when provided by a health care provider trained in abortion care?" Multiple responses were allowed, and response options were not read out loud. We explored several measures of knowledge. First, we descriptively examined responses, which included: if the woman's life is at risk; if the women's physical health is at risk; if pregnancy is from rape; if pregnancy is from incest; if the fetus would be handicapped/fetal anomaly; if the woman is mentally incapacitated; if the woman's mental health is at risk; under any circumstances; under no circumstances; and other. In our analysis, we created binary variables for each indication (yes/no) if respondents correctly identified that legal indication.

We grouped the responses that respondents stated spontaneously into four categories to align with the way they are written in the Ghanian abortion law, although we classified "mental

health" into a separate outcome due to its broader applicability per the Ghana Health Service's 2012 Comprehensive Abortion Care Services Standards and Protocols [28]. We created binary outcome variables for each category (yes/no) if respondents correctly identified all legal indications defining the category:

1. Pregnancy involves risk to life or physical health ("life/health")

2. Pregnancy involves risk to mental health ("mental health")

3. When there is substantial risk of fetal anomaly ("fetal anomaly")

4. Where pregnancy is the result of rape, incest or the woman is mentally incapacitated (written in the law as "defilement of the female idiot") ("rape/incest/defilement")

We also explored the number of existing legal conditions known by respondents, classifying "complete knowledge" as knowing all seven legal indications. Those who said abortion was not legal for any indication or who replied "don't know", were classified as knowing zero legal conditions. We explored write-in responses and grouped them into existing categories when appropriate, and included descriptive statistics for write-in responses that appeared repeatedly (e.g. abortion being legal for minors). Respondents who said abortion was legal on request or for "any" indication were classified as "overestimating" legality. We explored the proportion of respondents who perceived abortion as legal for fewer indications than it currently is (i.e., underestimating legality) relative to those who believed abortion to be as or more legal than it is (i.e. accurately or overestimating legality). We grouped those who accurately or overestimated into one category, since we assumed their perceptions of legality would be positively associated with provision, relative to those who underestimated the legality of abortion.

An additional outcome was respondents' recommendations for reducing unsafe abortion, assessed via the question "Please mention any suggestions / recommendations that you feel could be used in your region to reduce the number of unsafe abortions and their consequences for women's health." Multiple responses were allowed and response options were not read out loud. We examined three different binary outcomes: (1) educate the public on the abortion law and services (yes/no), (2) publicize the health risk involved in unsafe abortion (yes/no), and (3) improve the coverage and quality of comprehensive abortion care (CAC) services (yes/no). We examined responses among those who made recommendations and excluded two respondents who made no suggestions.

### Independent variables

We considered a range of respondent- and facility-level characteristics as independent variables. Respondent-level covariates included age (19-29/30-39/40+), gender (male/female), position (physician, midlevel provider, other) and years in current occupation (0–5 years/6 + years). Facility-level covariates included facility level (primary vs. secondary/tertiary), sector (public/private), location (urban/rural), and provision of post-abortion care versus provision of post-abortion care and/or induced abortion.

### Analysis

We first conducted descriptive analyses of our various measures of knowledge of legality and of our sample characteristics at the respondent and facility levels. We used bivariate and multi-variable logistic regression models to explore relationships between respondent- and facility-level factors and binary knowledge indicators including (1) knowledge of four separate areas of the law (life/ health, rape/incest/defilement, fetal anomaly, and mental health), (2) the underestimation of abortion legality indicator (i.e. identifying zero to six legal indications) and

(3) the three measures of respondent recommendations for reducing unsafe abortion (educate the public, publicize risks, and improve coverage and quality). As age and years in current occupation were strongly correlated (correlation coefficient of 0.61), we removed age and kept years in current occupation in the multivariate model to be able to assess the influence of job training on legal knowledge and recommendations.

We used likelihood ratio tests to assess the contribution of predictor variables to each model, and Hosmer-Lemeshow tests to assess goodness of fit of our final models. We also tested for multicollinearity using variance inflation factors (VIF); none of our variables had a VIF higher than 2, signaling that multicollinearity was not of concern. Analyses were weighted to generate representative estimates, and weights were calculated as non-response-adjusted inverses of selection probabilities for each sample facility [7]. We analyzed data using Stata 16.1 (Statcorp LP, College Station, TX).

### Inclusivity in global research

Further detail on considerations related to inclusivity in global research are included in the appendix (S2 File).

## Results

### Sample characteristics

The characteristics of the 340 respondents, each representing a facility, are presented in Table 1. Nearly half (48.4%) of respondents were over the age of 40, and 66.3% were female. Most respondents (75.8%) were midlevel providers, while 19.3% were physicians. More than half (58.5%) had been in their occupation for more than six years. Half of respondents (55.9%) worked in primary-level facilities. There was a nearly even divide between public and private sector facilities (48.2% and 51.8% respectively), and slightly more facilities were in urban areas (52.4%). All facilities in our sample provided postabortion and/or induced abortion care: 64.7% provided PAC only, 34.2% provided both PAC and abortion, and 1.07% provided only induced abortion.

### Knowledge of legal conditions for abortion in Ghana

Complete knowledge of the abortion law (i.e. identification of all seven existing legal indications) was low, at 6.3% (Table 2). Nearly half (45.5%) of respondents stated one to three conditions of the law. Only 3.1% of respondents did not correctly state any legal conditions, while 1.5% stated that abortion was not legal under any circumstance. Most participants (71.5%) stated that abortion was legal in cases where a pregnancy threatens a woman's life. The next most stated indication was in case of rape (57.7%), followed by incest (47.8%) and physical health (47.7%). Knowledge of the mental health indication was lower, with just 28.8% stating this indication. Further, one-quarter (25.3%) of respondents stated that abortion was legal if a mentally incapacitated person was defiled. A vast majority of respondents (83.3%) underestimated the number of circumstances under which abortion is legal. A variety of reasons beyond the law were also included as write-in responses, including age, poverty and at the woman's request; the most common write-in response was that abortion was legal for minors/ students, which was stated by 7% of respondents.

Regarding recommendations for reducing unsafe abortion, most respondents (63.1%) stated that educating the public on the abortion law would be helpful, while 55.5% suggested improving service availability and publicizing the risks of unsafe abortion. Fewer respondents (36.4%) recommended improving comprehensive abortion care.

**Table 1. Characteristics at the individual- and facility-level among health facility staff in Ghana[1].**

| Individual-level characteristics | N | % |
|---|---|---|
| **Age** | | |
| 19–29 years | 46 | 12.70 |
| 30–39 years | 137 | 38.81 |
| >40 years | 157 | 48.49 |
| **Gender** | | |
| Male | 121 | 33.68 |
| Female | 219 | 66.32 |
| **Position** | | |
| Physician (OB, GP, Specialist) | 63 | 19.27 |
| Midlevel Provider (PA, Nurse, MW) | 261 | 75.75 |
| Other (Administrator, Other) | 16 | 4.98 |
| **Years in current occupation** | | |
| 0–5 | 153 | 41.47 |
| 6+ | 187 | 58.53 |
| *Facility-level characteristics* | N | % |
| **Respondent reported facility type** | | |
| Primary | 192 | 55.78 |
| Secondary/Tertiary | 148 | 44.22 |
| **Sector** | | |
| Public | 183 | 48.21 |
| Private | 157 | 51.79 |
| **Facility urban or rural** | | |
| Urban | 177 | 52.38 |
| Rural | 163 | 47.62 |
| **Provision of abortion** | | |
| PAC only | 219 | 64.71 |
| Abortion only | 3 | 1.07 |
| Both PAC and abortion | 118 | 34.23 |
| **Total** | **340** | |

[1]Percentages are weighted, numbers are unweighted.

[2] 18 of these facilities reported that they do provide induced abortion but did not provide any abortions in the past year. This measure excludes facilities that reported providing PAC, but not induced abortion.

## Factors associated with knowledge of legal conditions

Unadjusted and adjusted associations between respondent- and facility-level characteristics and knowledge of each category of legal indications (life/physical health, rape/incest/defilement, fetal anomaly, and mental health) are presented in Table 3A and 3B. In our models, provision of induced abortion care in addition to PAC was significantly associated with having knowledge of rape/incest/defilement and mental health indications. Respondents working in private facilities had significantly lower odds of knowing the legal indications for rape, incest, and defilement of a mentally incapacitated individual (OR: 0.37, 95% CI: 0.19–0.72) in our unadjusted model, and mental health (aOR: 0.51, 95% CI: 0.28–0.95) in our adjusted model, compared to respondents in public facilities. Respondents with six or more years of experience also had lower odds of knowing the legal indications for fetal anomaly (aOR: 0.55, 95% CI: 0.34–0.87) and mental health (aOR: 0.61, 95% CI 0.39–0.97), compared to their counterparts with five years or less of experience.

**Table 2. Percentage distribution of legal knowledge and recommendations among health facility staff in Ghana.**

| Knowledge of abortion legality* | N | % |
|---|---|---|
| **Knows all seven circumstances when abortion is legal** | | |
| No (i.e., knew 0–6 reasons or stated that any reason is legal) | 315 | 93.66 |
| Yes (i.e. complete and accurate knowledge of the law) | 25 | 6.34 |
| **Number of legal conditions known** | | |
| No conditions | 9 | 3.14 |
| 1–3 conditions | 143 | 45.55 |
| 4–6 conditions | 126 | 34.57 |
| 7 conditions | 25 | 6.36 |
| Abortion available for any reason/on request | 36 | 10.37 |
| **Knowledge of specific legal indications** | | |
| Life at risk | 251 | 71.46 |
| Rape | 205 | 57.63 |
| Incest | 176 | 47.79 |
| Physical health at risk | 170 | 47.68 |
| Fetal anomaly | 157 | 43.33 |
| Mental health at risk | 101 | 28.80 |
| Defilement of mentally incapacitated person | 93 | 25.29 |
| Any reason/ on request | 36 | 10.33 |
| Other (Miscellaneous write-ins) | 71 | 21.68 |
| Minors/ students (Write-in response) | 23 | 7.20 |
| Under no circumstances | 4 | 1.45 |
| **Under/overestimated legality** | | |
| Underestimates (knows fewer than all seven conditions of law) | 278 | 83.26 |
| Accurate knowledge | 25 | 6.36 |
| Overestimates (perceives abortion as more legal) | 36 | 10.37 |
| *Spontaneously mentioned suggestions for reducing unsafe abortion* | | |
| Educate public on laws/ services | 224 | 63.09 |
| Publicize unsafe abortion | 197 | 55.46 |
| Improve comprehensive abortion care | 134 | 36.41 |
| **Total** | **340** | |

* Data on legal knowledge were missing for one participant.

## Factors associated with underestimating the number of conditions under which abortion is legal

The adjusted model examining factors related to underestimating legal conditions for abortion shows that respondents working in facilities providing induced abortion care in addition to PAC were significantly less likely to underestimate legality than those working in facilities providing only PAC (aOR: 0.45, 95% CI: 0.26–0.79) (Table 4).

## Factors associated with recommendations for reducing unsafe abortion

Finally, we explored factors associated with various recommendations for reducing unsafe abortion (Table 5). In adjusted models, we found private sector staff were significantly less likely than public sector staff to recommend educating the public (aOR: 0.58, 95% CI: 0.33–0.99). Non-medical staff were less likely than physicians to recommend improving comprehensive abortion care (aOR: 0.22, 95% CI: 0.05–0.90), and those working in facilities providing

**Table 3. a. Factors associated with knowledge of the conditions of the law among health facility staff in Ghana[1].** b. Factors associated with knowledge of the conditions of the law among health facility staff in Ghana[1].

| | Life/health | | | | Rape/incest/defilement | | | |
|---|---|---|---|---|---|---|---|---|
| | OR | 95% CI | aOR | 95% CI | OR | 95% CI | aOR | 95% CI |
| **Gender** | | | | | | | | |
| Man | Ref | - | Ref | - | Ref | - | Ref | - |
| Woman | 1.37 | (0.90 - 2.09) | 1.43 | (0.83–2.48) | 0.93 | (0.58–1.51) | 0.75 | (0.42–1.32) |
| **Position** | | | | | | | | |
| Physician (OB, GP, Other specialist) | Ref | - | Ref | - | Ref | - | Ref | - |
| Midlevel provider (PA, Nurse, Midwife) | 0.86 | (0.51 - 1.44) | 0.87 | (0.44–1.72) | 1.06 | (0.53–2.13) | 0.79 | (0.33–1.92) |
| Other (Administrator, Other) | 0.57 | (0.20 - 1.60) | 0.65 | (0.19–2.20) | - | | - | |
| **Years in current occupation** | | | | | | | | |
| 0–5 | Ref | - | Ref | - | Ref | - | Ref | - |
| 6+ | **1.57** | **(1.02 - 2.41)** | 1.38 | (0.85–2.23) | 0.56 | (0.31–1.02) | 0.62 | (0.29–1.35) |
| **Sector** | | | | | | | | |
| Public | Ref | - | Ref | - | Ref | - | Ref | - |
| Private | 1.17 | (0.77 - 1.78) | 0.78 | (0.49–1.25) | **0.37** | **(0.19–0.72)** | 0.46 | (0.20–1.08) |
| **Facility level** | | | | | | | | |
| Primary | Ref | - | Ref | - | Ref | - | Ref | - |
| Secondary/Tertiary | **2.01** | **(1.36 - 2.99)** | **1.86** | **(1.18–2.92)** | 0.67 | (0.39–1.15) | 0.72 | (0.40–1.29) |
| **Facility urban or rural** | | | | | | | | |
| Urban | Ref | - | Ref | - | Ref | - | Ref | - |
| Rural | **0.50** | **(0.33 - 0.77)** | **0.61** | **(0.39–0.95)** | 0.95 | (0.56–1.59) | 0.69 | (0.40–1.21) |
| **Abortion provision[2]** | | | | | | | | |
| Provides PAC only | Ref | - | Ref | - | Ref | - | Ref | - |
| Provides abortion &/or PAC | 0.89 | (0.58 - 1.38) | 0.70 | (0.44–1.11) | **1.82** | **(1.01–3.27)** | 1.71 | (0.86–3.40) |
| | Fetal anomaly | | | | Mental health | | | |
| | OR | 95% CI | aOR | 95% CI | OR | 95% CI | aOR | 95% CI |
| **Gender** | | | | | | | | |
| Man | Ref | - | Ref | - | Ref | - | Ref | - |
| Woman | 0.68 | (0.43–1.07) | 0.67 | (0.39–1.15) | 0.93 | (0.59–1.45) | 1.06 | (0.57–1.97) |
| **Position** | | | | | | | | |
| Physician (OB, GP, Other specialist) | Ref | - | Ref | - | Ref | | Ref | - |
| Midlevel provider (PA, Nurse, Midwife) | 0.92 | (0.51–1.65) | 0.99 | (0.48–2.02) | 0.76 | (0.43–1.34) | 0.44 | (0.19–1.03) |
| Other (Administrator, Other) | 0.64 | (0.26–1.56) | 0.70 | (0.29–1.74) | 0.33 | (0.07–1.47) | 0.36 | (0.07–1.87) |
| **Years in current occupation** | | | | | | | | |
| 0–5 | Ref | - | Ref | - | Ref | - | Ref | - |
| 6+ | **0.52** | **(0.35–0.77)** | **0.55** | **(0.34–0.87)** | **0.63** | **(0.42–0.94)** | **0.61** | **(0.39–0.97)** |
| **Sector** | | | | | | | | |
| Public | Ref | - | Ref | - | Ref | - | Ref | - |
| Private | 0.83 | (0.53–1.30) | 0.86 | (0.49–1.49) | **0.45** | **(0.27–0.75)** | **0.51** | **(0.28–0.95)** |
| **Facility level** | | | | | | | | |
| Primary | Ref | - | Ref | - | Ref | - | Ref | - |
| Secondary/Tertiary | 0.77 | (0.52–1.14) | 0.87 | (0.56–1.35) | 0.75 | (0.50–1.13) | 0.75 | (0.44–1.29) |
| **Facility urban or rural** | | | | | | | | |
| Urban | Ref | - | Ref | - | Ref | - | Ref | - |
| Rural | 0.94 | (0.65–1.37) | 0.78 | (0.51–1.18) | 1.11 | (0.72–1.73) | 0.96 | (0.58–1.60) |
| **Abortion provision[2]** | | | | | | | | |
| Provides PAC only | Ref | - | Ref | - | Ref | | Ref | - |

**Table 3.** (Continued)

| | Life/health | | | | Rape/incest/defilement | | | |
|---|---|---|---|---|---|---|---|---|
| | OR | 95% CI | aOR | 95% CI | OR | 95% CI | aOR | 95% CI |
| Provides abortion &/or PAC | 0.75 | (0.49–1.13) | 0.80 | (0.50–1.29) | **2.25** | **(1.41–3.60)** | **2.10** | **(1.30–3.38)** |

[1] Bolded estimates are statistically significant at p<0.05.

[2] This variable distinguishes those who provided abortion and/or PAC versus those that provided only PAC but not abortion.

OR: Odds Ratio; CI: Confidence Interval.

both abortion and PAC were more likely to recommend educating the public on the abortion law than those in facilities providing just PAC (aOR: 1.87, 95% CI: 1.10–3.16). Female health facility staff were less likely than male staff to recommend educating the public on the abortion law (aOR: 0.50, 95% CI: 0.28–0.90), but more likely than men to recommend publicizing the risks of unsafe abortion (aOR: 1.76, 95% CI: 1.02–3.05). Finally, those working in rural areas were significantly less likely to recommend educating the public on the abortion law compared to those in urban areas (aOR: 0.59, 95% CI: 0.35–0.97).

**Table 4. Factors associated with underestimating the legality of abortion among health facility staff in Ghana[1].**

| | OR | 95% CI | aOR | 95% CI |
|---|---|---|---|---|
| **Gender** | | | | |
| Man | Ref | - | Ref | - |
| Woman | 1.23 | (0.75–2.03) | 1.21 | (0.70–2.10) |
| **Position** | | | | |
| Physician (OB, GP, Other specialist) | Ref | - | Ref | - |
| Midlevel provider (PA, Nurse, Midwife) | 1.23 | (0.65–2.33) | 1.32 | (0.58–3.00) |
| Other (Administrator, Other) | 3.31 | (0.47–23.14) | 3.21 | (0.42–24.45) |
| **Years in current occupation** | | | | |
| 0–5 | Ref | - | Ref | - |
| 6+ | 1.40 | (0.76–2.59) | 1.46 | (0.71–2.96) |
| **Sector** | | | | |
| Public | Ref | - | Ref | - |
| Private | 1.42 | (0.84–2.41) | 1.03 | (0.54–1.97) |
| **Facility level** | | | | |
| Primary | Ref | - | Ref | - |
| Secondary/Tertiary | 1.14 | (0.65–2.00) | 1.14 | (0.59–2.19) |
| **Facility urban or rural** | | | | |
| Urban | Ref | - | Ref | - |
| Rural | 0.72 | (0.45–1.14) | 0.75 | (0.44–1.26) |
| **Abortion provision[2]** | | | | |
| Provides PAC only | Ref | - | Ref | - |
| Provides abortion &/or PAC | **0.48** | **(0.27–0.84)** | **0.45** | **(0.26–0.79)** |

[1] Bolded estimates are statistically significant at p≤0.05.

[2] This variable distinguishes those who provided abortion and/or PAC versus those that provided only PAC but not abortion.

OR: Odds Ratio; CI: Confidence Interval.

**Table 5. Factors associated with making recommendations for reducing unsafe abortion in Ghana[1].**

| | Educate public on laws | | | | Publicize risks of unsafe abortion | | | | Improve comprehensive abortion care | | | |
|---|---|---|---|---|---|---|---|---|---|---|---|---|
| | OR | 95% CI | aOR | 95% CI | OR | 95% CI | aOR | 95% CI | OR | 95% CI | aOR | 95% CI |
| **Gender** | | | | | | | | | | | | |
| Man | Ref | - | Ref | - | Ref | - | Ref | - | Ref | - | Ref | - |
| Woman | 0.76 | (0.49–1.19) | **0.50** | **(0.28–0.90)** | **1.73** | **(1.10–2.74)** | **1.76** | **(1.02–3.05)** | 0.86 | (0.58–1.29) | 1.04 | (0.61–1.76) |
| **Position** | | | | | | | | | | | | |
| Physician (OB, GP, Other specialist) | Ref | - | Ref | - | Ref | - | Ref | - | Ref | - | Ref | - |
| Midlevel provider (PA, Nurse, Midwife) | 1.20 | (0.73–1.98) | 1.66 | (0.82–3.38) | **1.83** | **(1.07–3.13)** | 0.99 | (0.48–2.02) | 0.75 | (0.43–1.31) | 0.51 | (0.23–1.10) |
| Other (Administrator, Other) | 1.16 | (0.37–3.70) | 2.00 | (0.59–6.80) | 2.26 | (0.76–6.73) | 1.88 | (0.59–6.02) | **0.24** | **(0.06–0.94)** | **0.22** | **(0.05–0.90)** |
| **Years in current occupation** | | | | | | | | | | | | |
| 0–5 | Ref | - | Ref | - | Ref | - | Ref | - | Ref | - | Ref | - |
| 6+ | 0.87 | (0.57–1.33) | 1.03 | (0.62–1.69) | **0.64** | **(0.44–0.92)** | 0.67 | (0.44–1.04) | 0.81 | (0.56–1.16) | 0.91 | (0.62–1.35) |
| **Sector** | | | | | | | | | | | | |
| Public | Ref | - | Ref | - | Ref | - | Ref | - | Ref | - | Ref | - |
| Private | **0.51** | **(0.33–0.81)** | **0.58** | **(0.33–0.99)** | **0.67** | **(0.45–1.00)** | 0.73 | (0.44–1.21) | **0.63** | **(0.41–0.97)** | 0.63 | (0.39–1.03) |
| **Facility level** | | | | | | | | | | | | |
| Primary | Ref | - | Ref | - | Ref | - | Ref | - | Ref | - | Ref | - |
| Secondary/Tertiary | **0.65** | **(0.44–0.96)** | 0.65 | (0.40–1.07) | 0.87 | (0.56–1.36) | 1.07 | (0.63–1.79) | 0.80 | (0.55–1.16) | 0.90 | (0.59–1.40) |
| **Facility urban or rural** | | | | | | | | | | | | |
| Urban | Ref | - | Ref | - | Ref | - | Ref | - | Ref | - | Ref | - |
| Rural | 0.83 | (0.50–1.37) | **0.59** | **(0.35–0.97)** | 1.16 | (0.80–1.69) | 1.01 | (0.68–1.50) | 1.37 | (0.87–2.16) | 1.40 | (0.88–2.20) |
| **Abortion provision[2]** | | | | | | | | | | | | |
| Provides PAC only | Ref | - | Ref | - | Ref | - | Ref | - | Ref | - | Ref | - |
| Provides abortion &/or PAC | 1.85 | (1.16–2.95) | **1.87** | **(1.10–3.16)** | 0.86 | (0.55–1.34) | 0.78 | (0.49–1.24) | 1.13 | (0.72–1.77) | 0.97 | (0.61–1.56) |

[1] Bolded estimates are statistically significant at p<0.05.

[2] This variable distinguishes those who provided abortion and/or PAC versus those that provided only PAC but not abortion.

OR: Odds Ratio; CI: Confidence Interval.

## Discussion

We found that complete and accurate knowledge of the legal conditions permitting abortion in Ghana was low among staff in health facilities offering abortion or post-abortion care, with considerable variation across conditions: while most respondents knew some of the circumstances under which abortion is legal, few had comprehensive knowledge of the legal conditions, and 83% underestimated the number of conditions for which abortion is legal. Knowledge was considerably higher for more restrictive indications (e.g., if a woman's life is at risk) while other, more broadly interpretable indications (e.g., mental health) were less known. Those working in facilities that provided induced abortion care in addition to PAC often had better knowledge of the legal indications for abortion than those working in facilities providing only PAC, and were less likely to underestimate abortion legality; this could reflect exposure to a wider range of clients seeking abortion care for different legal indications. Those working in the public sector and with more recent training often had better knowledge of abortion legality

as well. These results highlight the importance of the widespread dissemination of the Comprehensive Abortion Care Services Standards and Protocols, as well as continuing medical education to increase the number of facilities and providers willing to offer induced abortion care to the fullest extent of the law.

Few nationally representative studies have examined knowledge of the national abortion law among providers and health facility staff in facilities that offer abortion. In their nationally representative health facility survey in Zimbabwe, Madziyire et al. (2019) found that 25% of health care providers knew all the conditions under which abortion is legal, significantly more than in our study. Considerably more respondents in Zimbabwe than in our study knew that abortion was legal in cases of rape and when a woman's physical health is at risk. Similar to our findings, roughly half of providers in Zimbabwe were aware abortion was legal in cases of incest and fetal anomaly [18]. While not directly comparable to our estimates among health facility staff, several small quantitative and qualitative studies have examined knowledge of the law among providers in Ghana. Similar to a study among physicians in a teaching hospital in Kumasi, just 3% of respondents in our survey did not know any legal indications for abortion [30]. Yet while 54% of surveyed physicians in Kumasi thought abortion was legal to save a woman's life or preserve her physical and mental health [30], we found considerable variability in knowledge of these indications, with 71%, 47% and 29% of respondents stating that abortion is legal if a woman's life, physical or mental health is at risk, respectively.

Limited and variable knowledge of the abortion law among providers and facility staff could be impeding access to legal abortion care, since people seeking abortion care often rely on facility staff for information, making them gatekeepers to access [13]. In a qualitative study with Ob/Gyns, nurse-midwives, and pharmacists in Accra, some providers who perceived abortion as being illegal were unwilling to provide abortion care due to concerns of legal risk. In particular, the law's specification that only certain providers (i.e. registered medical practitioners) can legally offer abortion care was a deterrent to some who were unclear as to whether they were included in that provision, which is primarily interpreted as including only medical doctors [25]. Importantly, the Ghana Health Service's 2006 Comprehensive Abortion Care Services Standards and Protocols clarified that doctors, obstetricians, nurse-midwives, midwifery-trained community health officers and medical assistants could all legally offer abortion care, with varying restrictions by gestational age, facility level, and type of procedure; however, wider dissemination of these guidelines appears necessary to improve awareness and implementation [25,27]. In a cross-sectional survey of 213 health practitioners in northern Ghana, knowledge of the abortion law was considerably lower among those who identified as "conscientious objectors" (i.e. refused to provide abortion due to moral or religious beliefs) [31]. Beyond Ghana, Kung et al. (2018) examined access to abortion under the health exception in Britain, Colombia and Mexico and found that knowledge of the law among physicians was a key factor in how well health exceptions were applied in practice. Variability in knowledge of health exceptions is thus a concern for access to legal abortion care in Ghana. In some countries with similar legal indications, particularly relating to mental health, the law has been broadly applied so as to expand access to safe abortion care, such as in New Zealand [14]. Thus, ensuring accurate knowledge of the law among providers and facility staff, particularly of health exceptions, is imperative for improving access to safe abortion services and reducing abortion-related complications in Ghana.

The Ministry of Health and the Ghana Health Service have made efforts to improve access to safe abortion care, including by expanding the types of providers that can offer such care, and allowing for broad interpretation of the law to increase access to safe abortion care (the country's comprehensive abortion care guidelines specifically allow for broad interpretation of the health indication) [32]. Thus, continuous medical training which includes information on

the abortion law and its possible interpretation, as well as diffusion of evidence-based clinical guidelines, is essential for continuing efforts to expand safe abortion access in Ghana. Training and hiring staff and medical providers who are willing to support and/or provide abortion care to the fullest extent of the law is also imperative to abortion access. Even among providers who refuse to offer abortion care, knowledge of the law is important to facilitate appropriate referrals to other providers. Future research should explore knowledge of Comprehensive Abortion Care Services Standards and Protocols among providers, to identify places where further outreach and awareness are needed. Finally, beyond providers and facility staff, efforts are needed to inform the public about the abortion law, as recommended by the majority of our respondents. Research in Ghana and other African countries has shown that women's knowledge of abortion legality is often limited, which may impede their ability to advocate for their legal right to abortion care [8,33]. More funding is needed to support these efforts [32].

This research has a number of strengths. Our study provides the first nationally representative estimates of knowledge of abortion legality among a range of providers and staff in health facilities offering abortion or post-abortion care in Ghana, excluding NGO and community-based health planning and services (CHPS) facility personnel. We were able to explore physicians, midlevel providers, and administrators' knowledge of each condition of the abortion law which provides a more complete understanding of existing gaps in knowledge. However, our study has limitations. We were unable to assess knowledge of legality among staff from facilities that do not provide abortion or post-abortion care, who would likely have poorer knowledge of the law. Our sample excludes NGO and CHPS facilities, which comprise 24.9% of abortion-providing facilities in Ghana, and this may bias our estimates if knowledge among this population differs substantially. Our measure of knowledge of the law did not have response options read aloud, and it is possible that with probing on various legal indications, respondents' knowledge of the law would have appeared greater. We also decided to group some legal indications in our analysis (e.g., threat to life and health) to simplify the presentation of our results rather than present more precise regression results for each legal indication, which may impede the comparison of differences between the legal indications. Finally, we did not explore knowledge of the Comprehensive Abortion Care Services Standards and Protocols, which provide important information beyond the law to enable providers to know the circumstances under which they can provide legal abortion care.

## Conclusion

While many health facility staff in Ghana know some legal indications for abortion, complete knowledge of the abortion law is low, and most respondents underestimate the legality of abortion. Knowledge of indications with broader interpretability, i.e., mental health, was particularly low. Knowledge of the law among health providers and other facility staff is important for ensuring abortion care is made available to the fullest extent of the law in Ghana. Efforts are needed to improve knowledge of the law among providers and facility staff, particularly for health indications and their potentially broad interpretability.

## Supporting information

**S1 File. STROBE checklist.**
(DOCX)

**S2 File. Inclusivity in global research questionnaire.**
(DOCX)

## Author Contributions

**Conceptualization:** Grace Sheehy, Chelsea Polis, Caroline Moreau.

**Formal analysis:** Grace Sheehy.

**Investigation:** Chelsea Polis, Easmon Otupiri.

**Methodology:** Grace Sheehy, Chelsea Polis, Easmon Otupiri, Caroline Moreau.

**Project administration:** Chelsea Polis, Easmon Otupiri.

**Supervision:** Chelsea Polis, Easmon Otupiri, Caroline Moreau.

**Writing – original draft:** Grace Sheehy.

**Writing – review & editing:** Grace Sheehy, Chelsea Polis, Easmon Otupiri, Caroline Moreau.

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
