## [Decision Letter · Decision Letter 0]

26 Oct 2023

PONE-D-23-17111Knowledge of abortion legality among health facility staff in GhanaPLOS ONE

Dear Dr. Sheehy,

Thank you for submitting your manuscript to PLOS ONE. After careful consideration, we feel that it has merit but does not fully meet PLOS ONE’s publication criteria as it currently stands. Therefore, we invite you to submit a revised version of the manuscript that addresses the points raised during the review process.

Please note that we have only been able to secure a single reviewer to assess your manuscript. We are issuing a decision on your manuscript at this point to prevent further delays in the evaluation of your manuscript. Please be aware that the editor who handles your revised manuscript might find it necessary to invite additional reviewers to assess this work once the revised manuscript is submitted. However, we will aim to proceed on the basis of this single review if possible.

We look forward to receiving your revised manuscript.

Kind regards,

Jianhong Zhou

Staff Editor

PLOS ONE

Journal Requirements:

"This study was made possible by UK Aid from the UK Government (Component Code: 203177 -101 Purchase Order: 40054781) and a grant from the Dutch Ministry of Foreign Affairs (Activity #4000000282). The findings and conclusions contained within are those of the authors and do not necessarily reflect positions or policies of the donors."

Reviewers' comments:

Reviewer's Responses to Questions

**Comments to the Author**

1. Is the manuscript technically sound, and do the data support the conclusions?

Reviewer #1: Yes

2. Has the statistical analysis been performed appropriately and rigorously? 

Reviewer #1: I Don't Know

3. Have the authors made all data underlying the findings in their manuscript fully available?

Reviewer #1: Yes

4. Is the manuscript presented in an intelligible fashion and written in standard English?

Reviewer #1: Yes

5. Review Comments to the Author

Reviewer #1: This is a valuable contribution to the literature as the only study on nationwide knowledge of abortion law in Ghana. I do not have the background to offer comments on quantitative research methods. I offer the following:

- In explaining the law in the introduction Authors should cite and reference a primary source for the law, available through the GAPD (citation #1).

- The groundbreaking piece of the 2006 guidelines ("Standards and Protocols") (and in the 2012 update, available in the GAPD) was the expanded access under the health indications for abortion. Knowing the specifics words of the law itself would not enable providers to know the circumstances under which they can provide care, but rather the broad definition of health (or mental health?) (which includes WHO definition and social circumstances) in the Standards and Protocols. Authors should mention this in the introduction and consider throughout. Particularly the discussion in para 3 of the intro -- true abortion laws are written unclearly, and the S&Ps (in Ghana) are instrumental to addressing the ambiguity in the law. Nearly every abortion law sets out penalties then exceptions; this is not unique to Ghana (as described in FN 8). Authors could still cite the Ghana-specific publication as an example however.

- Introduction: Consider citing WHO abortion guidelines or human rights standards to give further weight to the need to provide abortion to the fullest extent of the law.

- Excellent overview of studies of abortion law knowledge, which will be a resource for others.

- Fn 15 only refers to a study in Ghana, not "studies have found" (as written it seems like multiple countries)

- Another place for reference to the S&Ps, study in FN 17 in fact references need for S&P dissemination to clear ambiguities in the law.

- Methods: It would be helpful to include a definition or description of what puts a facility in the category of "could potentially provide abortion". Is this from a legal standpoint? Or skill of available providers and equipment?

- p. 8: Mental health is not merely a possibly broader applicability; the S&Gs spell out its broad applicability.

- p. 21: Authors could consider adding specifics about the specification of "only certain providers", which appear in the law as MPs, while the S&Gs expand the cadres of providers.

Overall an important study and written very clearly. Given that the study focuses on knowledge of the law, authors should (as above and overall) familiarize themselves with and include specific provisions of the law and S&Ps, and citing primary sources.

6. PLOS authors have the option to publish the peer review history of their article (what does this mean?). If published, this will include your full peer review and any attached files.

Reviewer #1: **Yes: **Patty Skuster

---

## [Author Response · Author response to Decision Letter 0]

6 Feb 2024

PONE-D-23-17111

Knowledge of abortion legality among health facility staff in Ghana

PLOS ONE

January 31, 2024

Response to Editors

Dear PLOS ONE Editors,

Thank you very much for the opportunity to revise and resubmit our manuscript, Knowledge of abortion legality among health facility staff in Ghana. Below we have responded to each of the recommended revisions. We look forward to hearing back with a final decision on the manuscript.

Best wishes,

Grace Sheehy 

Notes to Editor:

In addition to the revisions requested by the reviewer, addressed below, we have made the following additions and modifications noted in the cover letter:

• We have provided a revised data availability statement which includes a link to the dataset and code.

• We have revised the role of funder statement.

• References: We have corrected the citation for one paper in our reference list that was erroneously linked to a paper that had been withdrawn. We also added several references to respond to reviewer feedback below.

• Tables: We updated the weights used in our analysis which resulted in minor changes to decimal points in several tables, which are all noted in track changes. This did not result in any significant changes to our results.

• File names: All file names have been updated in accordance with PLOS’s formatting policies. 

Response to Reviewers

Dear Reviewer 1: Thank you very much for your time spent reviewing our manuscript and your thoughtful feedback. The suggestions made are much appreciated, and we have responded to each recommendation below. 

Reviewer #1: This is a valuable contribution to the literature as the only study on nationwide knowledge of abortion law in Ghana. I do not have the background to offer comments on quantitative research methods. I offer the following:

- In explaining the law in the introduction Authors should cite and reference a primary source for the law, available through the GAPD (citation #1).

Response to reviewer: Thank you for this suggestion, we have amended the references in the first paragraph of the introduction to include the primary source material (i.e. the 1960 Criminal Code and the 1985 Criminal Code amendment) where explanations of the law are provided.

- The groundbreaking piece of the 2006 guidelines ("Standards and Protocols") (and in the 2012 update, available in the GAPD) was the expanded access under the health indications for abortion. Knowing the specifics words of the law itself would not enable providers to know the circumstances under which they can provide care, but rather the broad definition of health (or mental health?) (which includes WHO definition and social circumstances) in the Standards and Protocols. Authors should mention this in the introduction and consider throughout. Particularly the discussion in para 3 of the intro -- true abortion laws are written unclearly, and the S&Ps (in Ghana) are instrumental to addressing the ambiguity in the law. Nearly every abortion law sets out penalties then exceptions; this is not unique to Ghana (as described in FN 8). Authors could still cite the Ghana-specific publication as an example however.

Response to reviewer: Thank you very much for this helpful comment and your insights on the importance of the S&Ps. We have made several revisions in the introduction and discussion sections to better address the importance of knowledge of the S&Ps among providers and have also noted a limitation of our study as not exploring knowledge of these guidelines as well. 

Our relevant modifications to the text are as follows:

• At the end of paragraph 3 (page 4) in the introduction, we added the following: In the absence of clear legal guidance, national guidelines, standards and protocols are often written to address this ambiguity and provide practical guidance for health facilities and providers [14]; however, widespread dissemination and uptake of these guidelines is imperative to ensure their utility. 

• At the end of paragraph 5 (page 6) in the introduction, we moved and expanded the following content: Martin and colleagues (2011) describe the ambiguity in Ghana’s abortion law and how this ambiguity functions as both a deterrent and a facilitator of abortion access – the Ghanaian law states that abortion is unlawful and a punishable offense, and then lists various circumstances under which abortion is legally permitted. This conflicting information creates a sense among some providers that abortion is illegal, breeding anxiety about the possibility of being prosecuted for providing abortion care and contributing to anti-abortion stigma, while offering others the sense that they have more latitude in providing care [25]. In 2006, the Ghana Health Service released guidance for addressing ambiguity in the abortion law via the Comprehensive Abortion Care Services Standards and Protocols, since updated in 2012 and again in 2021 [26–28]. The Standards and Protocols provide guiding principles on abortion care service implementation, defining mental health per the WHO definition as a “state of emotional, psychological and social wellbeing and not merely the absence of disease”, specifying that continuing a pregnancy “may put a client’s mental health at risk” and that no psychiatric assessment is required to meet this indication for a legal abortion [27]. However, wider dissemination of these guidelines is needed to clarify legal ambiguities among providers and health facility staff [24].

• We amended paragraph 1 of the discussion section (page 21) to read: These results highlight the importance of the widespread dissemination of the Comprehensive Abortion Care Services Standards and Protocols, as well as continuing medical education, to increase the number of facilities and providers willing to offer induced abortion care to the fullest extent of the law.

• On page 24, we also added: Future research should explore knowledge of the Comprehensive Abortion Care Services Standards and Protocols among providers, to identify places where further outreach and awareness are needed.

• And at the end of the discussion section (page 25), we added to our limitations: Finally, we did not explore knowledge of the Comprehensive Abortion Care Services Standards and Protocols, which provide important information beyond the law to enable providers to know the circumstances under which they can provide legal abortion care.

- Introduction: Consider citing WHO abortion guidelines or human rights standards to give further weight to the need to provide abortion to the fullest extent of the law.

Response to reviewer: Thank you for this recommendation, we have amended paragraph two of the introduction to add reference to WHO abortion guidelines and human rights standards to give weight to the need to provide abortion to the fullest extent of the law. We have referenced the WHO guidelines, the Universal Declaration of Human Rights, and the 2018 General comment No. 36 on article 6 of the International Covenant on Civil and Political Rights, on the right to life.

The new text at the bottom of page 3 reads: Ensuring abortion care is made available to the fullest extent of the law is a matter of human rights; denying access to legal abortion care constitutes a form of discrimination and a violation of the rights described in the Universal Declaration of Human Rights, including the rights to non-discrimination, life, health, privacy, and bodily autonomy, among others [9,10]. In 2022, the World Health Organization recommended the full decriminalization of abortion due to the adverse impacts of restrictive abortion laws [11].

- Excellent overview of studies of abortion law knowledge, which will be a resource for others.

Response to reviewer: Thank you! We’re glad you agree this will be a useful resource for others.

- Fn 15 only refers to a study in Ghana, not "studies have found" (as written it seems like multiple countries)

Response to reviewer: Thank you for flagging this discrepancy. We have added additional citations here from two studies in South Africa to further support this statement (references #21 and 22):

• Harries J, Stinson K, Orner P. Health care providers’ attitudes towards termination of pregnancy: A qualitative study in South Africa. BMC Public Health. 2009;9: 296. doi:10.1186/1471-2458-9-296

• Magwentshu M, Chingwende R, Jim A, van Rooyen J, Hajiyiannis H, Naidoo N, et al. Definitions, perspectives, and reasons for conscientious objection among healthcare workers, facility managers, and staff in South Africa: a qualitative study. Sexual and Reproductive Health Matters. 2023;31: 2184291. doi:10.1080/26410397.2023.2184291

- Another place for reference to the S&Ps, study in FN 17 in fact references need for S&P dissemination to clear ambiguities in the law.

Response to reviewer: Thank you for highlighting this, we have amended the text to include mention of the S&P dissemination recommendation.

The text on page 6 now reads: In 2006, the Ghana Health Service released guidance for addressing ambiguity in the abortion law via the Comprehensive Abortion Care Services Standards and Protocols, since updated in 2012 and again in 2021 [26–28]. The Standards and Protocols provide guiding principles on abortion care service implementation, defining mental health per the WHO definition as a “state of emotional, psychological and social wellbeing and not merely the absence of disease”, specifying that continuing a pregnancy “may put a client’s mental health at risk” and that no psychiatric assessment is required to meet this indication for a legal abortion [27]. However, wider dissemination of these guidelines is needed to clarify legal ambiguities among providers and health facility staff [24].

- Methods: It would be helpful to include a definition or description of what puts a facility in the category of "could potentially provide abortion". Is this from a legal standpoint? Or skill of available providers and equipment?

Response to reviewer: Thank you for this question, we have added further detail to this section to clarify what puts a facility in the category of “could potentially provide abortion”. 

The text on page 5 now reads: All facility types that could, in theory, provide abortion and/or post-abortion care (e.g., excluding highly specialized facilities such as eye clinics, nutrition centers, and district health directorates), except for community-based health planning and services (CHPS) facilities and non-governmental organizations (NGOs), were retained.

- p. 8: Mental health is not merely a possibly broader applicability; the S&Gs spell out its broad applicability.

Response to reviewer: Thank you for highlighting this point – we have removed the word “possible” and added reference to the 2012 S&Ps (additions in italics).

The text on page 9 now reads: “…we classified “mental health” into a separate outcome due to its broader applicability per the Ghana Health Service’s 2012 Comprehensive Abortion Care Services Standards and Protocols."

- p. 21: Authors could consider adding specifics about the specification of "only certain providers", which appear in the law as MPs, while the S&Gs expand the cadres of providers.

Response to reviewer: Thank you for this suggestion. We have modified the text to specify which providers are permitted to offer abortion care according to the abortion law, and the broader recommended cadres specified in the S&Ps (additions in italics).

The text on page 22 now reads: In particular, the law’s specification that only certain providers (i.e. registered medical practitioners) can legally offer abortion care was a deterrent to some who were unclear as to whether they were included in that provision, which is primarily interpreted as including only medical doctors [24]. Importantly, the Ghana Health Service’s 2006 Comprehensive Abortion Care Services Standards and Protocols clarified that doctors, obstetricians, nurse-midwives, and midwifery-trained community health officers and medical assistants could all legally offer abortion care, with varying restrictions by gestational age, facility level, and type of procedure; however, wider dissemination of these guidelines appears necessary to improve awareness and implementation [24,26]. 

Overall an important study and written very clearly. Given that the study focuses on knowledge of the law, authors should (as above and overall) familiarize themselves with and include specific provisions of the law and S&Ps, and citing primary sources.

Response to reviewer: Thank you again for your helpful feedback and for your time spent reviewing our paper!

---

## [Editor Report · Decision Letter 1]

2 May 2024

PONE-D-23-17111R1Knowledge of abortion legality among health facility staff in GhanaPLOS ONE

Dear Dr. Sheehy,

Thank you for submitting your manuscript to PLOS ONE. After careful consideration, we feel that it has merit but does not fully meet PLOS ONE’s publication criteria as it currently stands. Therefore, we invite you to submit a revised version of the manuscript that addresses the points raised during the review process.

Thank you for allowing me to review your initial revision.

It has been challenging to reach out to the initial reviewers for a second assessment of the paper.

Despite my efforts to request additional reviewers, all have declined without providing any reasons.

Upon reevaluation, I find your response to be thorough. However, I believe the introduction section could benefit from further enhancement, incorporating insights from the following papers:

1. Morhee, R. A. S., & Morhee, E. S. K. (2006). Overview of the law and availability of abortion services in Ghana. Ghana medical journal, 40(3).

2. Aniteye, P., & Mayhew, S. H. (2013). Shaping legal abortion provision in Ghana: using policy theory to understand provider-related obstacles to policy implementation. Health Research Policy and Systems, 11, 1-14.

3. Abubakari, A., Amankwa, A. M., Owusu, D., Fosuah, F., & Anane, J. Utilization of Comprehensive Abortion Care (CAC) and associated factors among young people (10-24 years): a cross-sectional study in the Tamale Metropolis. Northern Region of Ghana.

4. Abdul-Wahab, I., MubarickA., Nimota Nukpezah, R., & Kojo Dzantor, E. (2021). Adolescents' sexual and reproductive health: A survey of knowledge, attitudes, and practices in the Tamale Metropolis, Ghana.

I kindly request that you refine the introduction for clarity and conciseness.

While I understand the challenges of research, I encourage you to resubmit after improving the introduction section for final decision.

We look forward to receiving your revised manuscript.

Kind regards,

Mubarick Nungbaso Asumah, MPhil, Bsc

Academic Editor

PLOS ONE
---

## [Author Response · Author response to Decision Letter 1]

5 Jun 2024

Dear PLOS ONE Editors,

Thank you for the opportunity to revise and resubmit our manuscript, Knowledge of abortion legality among health facility staff in Ghana. Below we have responded to the recommended revisions. We look forward to hearing back with a final decision on the manuscript.

Best wishes,

Grace Sheehy 

Editor’s feedback: 

Upon reevaluation, I find your response to be thorough. However, I believe the introduction section could benefit from further enhancement, incorporating insights from the following papers:

1. Morhee, R. A. S., & Morhee, E. S. K. (2006). Overview of the law and availability of abortion services in Ghana. Ghana medical journal, 40(3).

2. Aniteye, P., & Mayhew, S. H. (2013). Shaping legal abortion provision in Ghana: using policy theory to understand provider-related obstacles to policy implementation. Health Research Policy and Systems, 11, 1-14.

3. Abubakari, A., Amankwa, A. M., Owusu, D., Fosuah, F., & Anane, J. Utilization of Comprehensive Abortion Care (CAC) and associated factors among young people (10-24 years): a cross-sectional study in the Tamale Metropolis. Northern Region of Ghana.

4. Abdul-Wahab, I., MubarickA., Nimota Nukpezah, R., & Kojo Dzantor, E. (2021). Adolescents' sexual and reproductive health: A survey of knowledge, attitudes, and practices in the Tamale Metropolis, Ghana.

I kindly request that you refine the introduction for clarity and conciseness.

Authors’ response to editor:

Thank you for your suggestions for strengthening the introduction section. We have incorporated three of the proposed references into this section (Morhee & Morhee 2006; Aniteye & Mayhew 2013; Abubakari et al. 2023) and have made some revisions for clarity and conciseness throughout the introduction.

---

## [Editor Report · Decision Letter 2]

23 Jul 2024

Knowledge of abortion legality among health facility staff in Ghana

PONE-D-23-17111R2

Dear Dr. Sheehy,

We’re pleased to inform you that your manuscript has been judged scientifically suitable for publication and will be formally accepted for publication once it meets all outstanding technical requirements.

Kind regards,

Mubarick Nungbaso Asumah, MPhil, Bsc

Academic Editor

PLOS ONE

Additional Editor Comments (optional):

Although, this is okay for publication, Language enhancement would make the entire paper a nice one.
---

## [Editor Report · Acceptance letter]

13 Aug 2024

PONE-D-23-17111R2 

PLOS ONE

Dear Dr. Sheehy, 

I'm pleased to inform you that your manuscript has been deemed suitable for publication in PLOS ONE. Congratulations! Your manuscript is now being handed over to our production team.

Kind regards, 

on behalf of

Dr. Mubarick Nungbaso Asumah 

Academic Editor

PLOS ONE